# The Relation between Negative Automatic Thoughts and Psychological Inflexibility in Schizophrenia

**DOI:** 10.3390/jcm11030871

**Published:** 2022-02-07

**Authors:** Cosmin O. Popa, Adrian V. Rus, Wesley C. Lee, Cristiana Cojocaru, Alina Schenk, Vitalie Văcăraș, Peter Olah, Simona Mureșan, Simona Szasz, Cristina Bredicean

**Affiliations:** 1Department of Ethics and Social Sciences, George Emil Palade University of Medicine, Pharmacy, Sciences and Technology, 540142 Tirgu Mures, Romania; cosmin.popa@umfst.ro; 2Department of Social and Behavioral Sciences, Southwestern Christian University, Bethany, OK 73008, USA; adrian.rus@swcu.edu (A.V.R.); wesley.lee@swcu.edu (W.C.L.); 3The Doctoral School of Medicine and Pharmacy, George Emil Palade University of Medicine, Pharmacy, Sciences and Technology, 540142 Tirgu Mures, Romania; cristiana-manuela.cojocaru@umfst.ro (C.C.); schenkalina@yahoo.com (A.S.); 4Department of Clinical Neurosciences, Iuliu Hațieganu University of Medicine and Pharmacy, 400006 Cluj-Napoca, Romania; vitalievacaras.umf@gmail.com; 5Department of Medical Informatics and Biostatistics, George Emil Palade University of Medicine, Pharmacy, Sciences and Technology, 540142 Tirgu Mures, Romania; 6Department of Internal Medicine IV, George Emil Palade University of Medicine, Pharmacy, Sciences and Technology, 540142 Tirgu Mures, Romania; 7Department of Rheumatology, Physical and Rehabilitation Medicine, George Emil Palade University of Medicine, Pharmacy, Sciences and Technology, 540142 Tirgu Mures, Romania; szasz_fc@yahoo.com; 8Department of Neuroscience, “Victor Babeș” University of Medicine and Pharmacy, NEUROPSY-COG Center for Cognitive Research in Neuropsychiatric Pathology, 300041 Timișoara, Romania; brediceancristina@gmail.com; 9Psychiatric Compartment, “Dr. Victor Popescu” Emergency Military Clinical Hospital, 300080 Timișoara, Romania

**Keywords:** schizophrenia, negative automatic thoughts, psychological inflexibility, cognitive fusion, experiential avoidance, cognitive behavioral therapy

## Abstract

Background: Schizophrenia is one of the most severe disorders in the Diagnostic and Statistical Manual of Mental Disorders (DSM-5) spectrum. Negative automatic thoughts (NAT), cognitive fusion (CF), and experiential avoidance (EA), as part of psychological inflexibility (PI), can be considered important dysfunctional cognitive processes in schizophrenia. Methods: In the present study, two samples were included: a target group consisting of 41 people with schizophrenia (23 females; aged 44.98 ± 11.74), and a control group consisting of 40 individuals with end-stage chronic kidney disease (CKD) (27 males; aged 60.38 ± 9.14). Results: Differences were found between the two groups, with patients with schizophrenia showing an increased frequency of NAT, as well as higher levels of CF and EA (psychological inflexibility), compared to the control group. NAT were the mediator in the relation between the schizophrenia diagnosis and CF, as well as EA. Conclusion: Individuals with schizophrenia present a specific dysfunctional pattern of cognitive functioning, in which negative automatic thoughts represent a distinctive pathway to cognitive fusion and experiential avoidance.

## 1. Introduction

Schizophrenia is one of the most severe mental disorders given its multiple negative psychological and social consequences, which markedly impact diverse areas of functioning like family, career, and self-care, all these aspects contributing to a poor quality of life in these patients [1,2]. From a clinical perspective, schizophrenia is considered a chronic disorder that includes positive symptoms, such as delusional thinking, auditory and/or visual hallucinations, as well as negative symptoms, such as lack of motivation, concentration deficits, anhedonia, and social isolation [3,4]. These features may contribute to the reluctance of patients to seek specialized treatment along with the emergence of symptoms [5]. According to a recent meta-analysis, estimates of lifetime prevalence of schizophrenia fall between 0.31% to 0.70% in the general population, with a high incidence of negative symptoms reported in males, although the risk of developing schizophrenia remains the same for both sexes [6]. Beyond the diagnostic and clinical aspects, cognitive deficits and several cognitive biases that could influence and exacerbate patients’ mood and clinical symptoms have been described in schizophrenia. Further, patients with schizophrenia present different cognitive deficits, a diminished capacity to evoke verbal information, impaired information processing, poor working memory, low problem-solving abilities, dysfunctional attentional/vigilance function, as well as an incapacity to retain different information or use visual memory [7,8]. Since schizophrenia has a chronic nature, besides social cognition biases, cognitive impairments are also strongly correlated with a poor quality of life in these patients, with negative implications for social relationships, family, profession and other life areas [9,10]. In this regard, different models stemming from cognitive-behavioral therapy (CBT) have described a number of cognitive mechanisms involved in the psychopathology of psychosis, such as negative automatic thoughts (NAT), and psychological inflexibility (PI), which include cognitive fusion (CF), and experiential avoidance (EA) [11,12].

According to Beck’s cognitive model, it is not life events that generate dysfunctional emotions, but more precisely the individual’s interpretations of these events, such as the experience of NAT. NAT have been defined as cognitive processes that occur at a conscious or pre-conscious level, presenting irrational, rigid content that typically produces significant emotional discomfort, as a reflection of the individual’s negative cognitive triad (negative thoughts about the self, world, and future) [13,14,15]. In primary psychotic disorders (PPD), the experience of NAT has been associated with the experience of depression and anxiety, as well as with the experience of psychotic spectrum symptoms like delusions and hallucinations [11,16,17,18,19]. Additionally, Beck’s cognitive model in schizophrenia can be better explained through a typical situation from clinical practice. For example, arguing with a friend (event), beneath the effect of delusions, the individual thinks that this friend has been replaced with another person (NAT). This interpretation of the event makes the individual feel angry (negative emotion) and start looking for evidence to confirm that what he/she thinks is real (behavior) [17]. In this way, when individuals with psychoses begin to debate their own NAT during CBT sessions, questioning their content, and succeeding to consider that those thoughts are not an absolute truth, this approach can be considered a part of a healthy metacognitive process with beneficial effects [20,21]. These are important aspects with regards to the psychological treatment of schizophrenia, since it has been demonstrated that distorted metacognitions are important biases most often correlated with specific symptoms in psychosis/schizophrenia that need to be targeted during treatment [22]. 

Furthermore, PI represents a core model of psychopathology derived from acceptance and commitment therapy (ACT) that comprises six components: (1) dominance of the conceptualized past or future—limited self-knowledge, (2) cognitive fusion, (3) experiential avoidance, (4) attachment to the conceptualized self, (5) lack of values clarity/contact, and (6) unworkable action [23,24]. PI is characterized by a rigid action/reaction style in which individuals lose contact with the present moment (distraction), present high levels of thought control, avoid accepting their feelings and internal experiences, and begin to identify with their problem (“I’m my illness/problem”). Additionally, life values are unclearly defined, and dysfunctional behavioral coping strategies override functional ones. Consequently, PI has a negative impact on the well-being and quality of life of individuals with psychotic disorders [25,26]. Across the entire ACT model of psychopathology, CF and EA are the processes that were most often correlated with the clinical symptoms of schizophrenia [27,28]. CF is characterized by overt and covert behaviors being excessively determined by cognitions, which means that patients struggle with their thoughts, attempt to control them, and have difficulties finding alternative explanations to these thoughts [29]. For example, the presence of auditory hallucinations can determine whether the individual believes “the voices” telling him that he is “a knight” who must fight the forces of evil or otherwise he will be punished by divinity. In these conditions, due to CF, the individual may start to be dominated by these voices, and his overt and covert behavior may be influenced by his cognitions. The individual may begin to struggle with his thoughts in the first instance, then be overpowered by them and create defensive strategies to be able to fight against the forces of evil. Consequently, he may engage in religious rituals and the transmission of mystical delusional messages on social media, and search for evidence to support his hypotheses. Additionally, he may present psychomotor agitation (overt behaviors), feel anxious, and/or depressed (covert behavior) [30,31]. According to the ACT theory, CF is interconnected with EA, in the sense that CF influences and increases the level of EA [32,33]. EA constitutes the most problematic psychological process from the entire spectrum of the PI model [24,32], since individuals tend to avoid their thoughts, feelings, physical sensations, memories, and so on. Therefore, this psychological mechanism can be considered the main factor involved in the long-term development of dysfunctional behaviors (e.g., social isolation, reassurance in anxiety, maintenance of maladaptive behaviors) [23]. EA has been associated with both positive and negative symptoms in schizophrenia, from the beginning and throughout different stages of the disorder, serving as a dysfunctional coping strategy that negatively influences quality of life and recovery [34,35]. Thus, NAT, EA and CF are associated with positive and negative symptoms in schizophrenia and represent cognitive factors that contribute to the maintenance and aggravation of these clinical symptoms.

Interestingly, cognitive processes like NAT and PI were also explored in previous studies focusing on organic/physical illnesses. From this perspective, there are many similarities between schizophrenia and physical diseases like end-stage chronic kidney disease (CKD), chronic pain, and other lifelong medical conditions [36,37,38]. Beyond the chronic aspects of these diseases and the associated poor quality of life, an important resemblance between schizophrenia and end-stage CKD is represented by the presence of PI and cognitive dysfunctions in both disorders. Particularly, individuals with end-stage CKD who require dialysis treatment present very high levels of PI and NAT, which are associated with depressive episodes and fatigue [39]. Given the association between the chronic aspects of end-stage CKD and their clinical implications, including multiple hospitalizations, dialysis schedule, and impaired professional, familial, social and role functioning, these patients develop multiple psychological problems, like depression and insomnia [40]. In addition, some of the most common cognitive dysfunctions in these patients are attentional problems. Particularly, a discrepancy between low IQ scores and intellectual performance regarding the verbal scores is another cognitive process involved in end-stage CKD [41]. Nevertheless, dialysis patients (end-stage CKD) have a poorer quality of life as compared with non-dialysis patients with CKD [42].

Consequently, there are some analogies, although with different clinical specificity, between schizophrenia and end-stage CKD dialyzed individuals, consisting in the chronicity of both conditions, the poor life quality, and the presence of NAT and PI at very high levels, as well as the occurrence of cognitive dysfunction.

In this context, this study is the first to investigate specific cognitive deficits in individuals with schizophrenia, as compared to individuals with end-stage CKD undergoing dialysis treatment. More specific, we focused on deficits related to NAT, EA and CF. The main objective of this study is the identification of differences in these cognitive processes involved in schizophrenia compared to an end-stage CKD group. The assumptions of this research are that (1) there are significant differences in NAT, CF and EA between individuals with schizophrenia when compared to an end-stage CKD group, and that (2) NAT represent a specific pathway to CF and EA in individuals with schizophrenia.

## 2. Materials and Methods

### 2.1. Participants and Procedure

The participants included in this study were recruited using a simple randomization method, from the Mures County Hospital, Psychiatric Clinic No. 2 in Tirgu Mures, Romania, and B. Braun Avitum Dialysis Center Clinic in Sighisoara, Mures, Romania. The study was carried out between 29 October 2020 and 1 September 2021 and was approved by the Institutional Ethics Committee of GE Palade University of Medicine, Pharmacy, Sciences and Technology in Tirgu Mures under the number 1166/29 October 2020. Participants received information regarding the research and signed the written consent, ensuring confidentiality and protection of data. The study comprised two groups and implemented a between-subjects design. The target group included 41 individuals (aged 44.98 ± 11.74), 23 females and 18 males, who had been diagnosed with schizophrenia. The control group comprised 40 individuals with end-stage CKD (aged 60.38 ± 9.14), 13 females and 27 males, without any psychiatric diagnosis. Given that previous studies focused on the differences between schizophrenia patients and non-psychiatric individuals [43,44], we had a different approach, investigating possible differences between various psychological processes involved in a chronic mental illness (i.e., schizophrenia) in comparison to a chronic physical condition (end-stage CKD) for the first time. The main reason for using these specific samples was that they share a common point represented by the presence of a persistent medical condition, the low welfare of patients and high levels of PI and NAT, despite the fact that the dysfunctional psychological processes could be higher in the schizophrenia group.

For the target group, the inclusion criteria were: (1) the existence of a schizophrenia diagnosis according to the criteria stated in the DSM-5; (2) hospitalization in a psychiatric clinic; (3) a time interval of at least five years from the onset of a schizophrenia diagnosis. The exclusion criteria were the following diagnoses: (1) first-episode psychosis (FEP), (2) substance-/medication-induced psychotic disorder (SMIPD), and (3) psychotic disorder due to another medical condition (PDDAMC). 

For the control group, the inclusion criteria were: (1) the diagnosis of end-stage CKD, as established by a medical doctor specialized in nephrology; (2) hemodialysis treatment; (3) the absence of any psychotic symptoms meeting the DSM-5 criteria. The exclusion criteria were: (1) the presence of any psychotic symptoms according to the DSM-5 criteria and (2) the presence of severe personality disorders, as established according to the intake interview conducted by a psychologist at the dialysis center. 

This clinical assessment was performed by a team of psychiatrists and clinical psychologists, using a standard psychiatric intake interview and the Structured Clinical Interview for DSM-5 (SCID-5-Clinical Version) [45], Module B—Psychotic and Associated Symptoms. The intake psychiatric interview represented the screening for identifying the possible presence of severe personality disorders. The first phase of the study also included a screening for establishing the diagnosis during which the section regarding schizophrenia was covered. This screening was performed by a psychiatrist at the first consultation, when the patients came for treatment/hospitalization, and consisted of a standard psychiatric intake interview and the application of the Structured Clinical Interview for DSM-5 (SCID-5-Clinical Version). Patients who met the conditions for the schizophrenia diagnosis were included in the target group, based on the following steps: (1) the informed consent was presented and signed by the patients, (2) a clinical intake interview for deciding the presence/absence of the schizophrenia diagnostic was performed, (3) the Structured Clinical Interview for DSM-5 (SCID-5-CV) was applied, and (4) the final diagnosis was established by correlating the conclusions of the psychiatric intake interview with those reported after the completion of the SCID-5-CV. After the informed consent was signed, a clinical assessment was also completed with participants in the control group, using both the clinical intake interview and the Structured Clinical Interview for the DSM-5 (SCID-5-CV), Module B—Psychotic and Associated Symptoms. In the end, individuals with no psychotic symptoms were enrolled in the control group. The second phase of this study featured an evaluation that involved the application of several instruments for measuring cognitive functioning, emotional variables, and patients’ psychological functioning. The psychological assessment took place one week after the initial interviews for group admission, and both groups were administered the Automatic Thoughts Questionnaire (ATQ) [46], the Cognitive Fusion Questionnaire (CFQ) [29] and the Acceptance and Action Questionnaire II (AAQ-II) [47]. Investigations were conducted by a team composed of psychiatrists and clinical psychologists who were trained to perform the psychiatric and psychological assessments, as well as to offer instructions to patients on how to complete the self-rating questionnaires. Regarding the dropout of the study, out of 85 patients diagnosed with schizophrenia who were offered to join the study, 24 refused, 12 only went through the first phase of the study consisting of the initial evaluation, and 8 were excluded because they did not complete the psychological self-report scales entirely. Some important conditions were that patients had to understand the Romanian language and the questionnaire items, as well as the conditions stated in the informed consent, such as the right to withdraw from the research anytime without adverse implications.

### 2.2. Measures

The Structured Clinical Interview for the DSM-5—Clinician Version (SCID-5-CV) [45] is a semi-structured interview for the assessment of DSM-5 psychopathology, evaluating both current and lifetime diagnoses for most disorders. It can be applied to a wide range of adult populations, including psychiatric and medical patients or individuals that do not report clinical symptoms. The SCID-5-CV requires the interviewer to establish the presence or absence of diagnostic criteria, asking additional clarifying questions if necessary. The instrument has been proven to have excellent reliability, with Kappa coefficients above 0.75 obtained for most diagnoses. Additionally, the sensitivity and specificity of the interview has been proven to be high, exceeding 0.70 and 0.80, respectively [48]. The structured clinical interview has been successfully applied to the Romanian population in a sample of patients with a psychiatric diagnosis, namely generalized anxiety disorder [49]. The present study used the Romanian version of SCID-5-CV, Module B—Psychotic and Associated Symptoms.

The Automatic Thoughts Questionnaire (ATQ) [46] is a 15-item self-report questionnaire that measures common negative self-statements or cognitions. The original version comprises 30 items measuring the type of thought contents particularly relevant in depression, such as “I’m worthless” and “My future is bleak”. Besides evaluating the frequency of these thoughts, the modified version of the ATQ also measures their credibility [50]. Respondents are asked to rate their answers using a Likert scale from 1 = not at all to 5 = all the time for frequency and 5 = totally for credibility. The total score is the sum of all ratings. Both the full and the shortened versions of the instrument have demonstrated excellent internal consistency, with Cronbach’s alpha coefficients exceeding 0.90 in a general population sample [51]. The scale has been used and adapted for Romania, with an obtained alpha Cronbach’s coefficient of 0.92 obtained in a general population sample [52,53]. 

The Cognitive Fusion Questionnaire (CFQ) [29] is a self-report measure of cognitive fusion, which refers to the tendency to consider that personal thoughts are literally true and to behave accordingly, despite the associated negative consequences for psychological functioning. The CFQ has 7 items that are rated on a 7-point Likert scale (from 1 = never true to 7 = always true). The total score is calculated by summing all items, with higher scores indicating increased cognitive fusion levels. The following item examples reflect the content of the scale: “I struggle with my thoughts”, “I get upset with myself for having certain thoughts”, and “I tend to get very entangled in my thoughts”. The CFQ has presented good test–retest reliability and excellent internal consistency, with Cronbach’s alpha levels around 0.90 obtained in the original validation study, which used multiple samples including both non-clinical and clinical populations with mental health problems like major depressive disorder and somatic conditions like multiple sclerosis [29]. The instrument has also been implemented cross-culturally, providing very good results in terms of psychometric properties, with a reliability alpha Cronbach’s coefficient of 0.96 obtained in a sample of Hispanic undergraduates [54]. The CFQ is currently in the process of validation for the Romanian population.

The Acceptance and Action Questionnaire II (AAQ-II) [47] is a self-report instrument developed for the assessment of psychological inflexibility, namely the propensity to avoid difficult internal experiences (i.e., experiential avoidance). The instrument has 7 items, and respondents are asked to rate their answers on a 7-point Likert scale (from 1 = never true to 7 = always true). The total score is calculated by adding up the responses for each item, with higher scores signifying increased psychological inflexibility. Several item examples can describe the scale content, such as “I’m afraid of my feelings” or “Worries get in the way of my success”. The AAQ-II is a reliable instrument, with a mean Cronbach’s alpha coefficient of 0.84 having been obtained when it was applied in diverse samples comprising people from the general population and who present problematic patterns of substance use [47]. The AAQ-II has been adapted for the Romanian population, resulting in a comparable internal consistency level, with a Cronbach’s alpha of 0.80 obtained in a young, non-clinical adult sample [55]. 

### 2.3. Data Analysis

First, the means (M), standard deviations (SD) and correlations (Pearson’s *r*) among the study variables were calculated. Second, to examine group differences between individuals with schizophrenia and individuals with end-stage CKD related to their NAT, EA, and CF, a multivariate analysis of variance (MANOVA) was employed. Finally, to examine the mediator role of NAT in the relation between groups (individuals with schizophrenia and individuals with end-stage CKD) and EA, as well as CF, we conducted two mediation analyses. We used a bootstrapping procedure (bias-corrected, with 5000 iterations) that assessed indirect effects. In this case, mediation occurs if the 95% bootstrapping confidence interval (CI) does not contain zero. All analyses were conducted using IBM SPSS Statistics, version 20 (IBM Corp., Armonk, NY, USA).

## 3. Results

### 3.1. Demographics

The demographic characteristics of individuals with schizophrenia and individuals with end-stage CKD are summarized in Table 1. The results showed significant differences between individuals with schizophrenia and with end-stage CKD with respect to gender (χ^2^ (1, *n* = 81) = 4.56, *p* = 0.033) and age (t (79) = 6.57, *p* < 0.001)). Individuals with schizophrenia were lower in age than those with end-stage CKD (mean difference = 15.39, 95% CI (10.73; 20.06)) and predominantly females (23 females and 18 males—schizophrenia group; 13 females and 27 males in the group without schizophrenia). 

### 3.2. Preliminary Analyses

Means, standard deviations, and Pearson’s r correlations among the study variables are shown in Table 2, Table 3 and Table 4. 

### 3.3. Differences between Individuals with Schizophrenia and Individuals with End-Stage CKD

A MANOVA was conducted to investigate differences between individuals with schizophrenia and individuals with end-stage CKD in their levels of NAT, EA, and CF. A significant difference emerged between the two groups (Wilks’ Lambda = 0.85, F (4, 76) = 3.192, *p* < 0.001). Individuals with schizophrenia showed an increased level of NAT (F (1, 79) = 9.833; *p* < 0.05, partial η^2^ = 0.111), more credibility in their NAT (F (1, 79) = 4.348; *p* < 0.05, partial η^2^ = 0.052), more EA (F (1, 79) = 6.775; *p* < 0.05, partial η^2^ = 0.079), and more CF (F (1, 79) = 6.834; *p* < 0.05, partial η^2^ = 0.080) in comparison to individuals with end-stage CKD (for more details, see Table 2). It is important to note that when age and gender were entered as covariates, the above-mentioned differences were no longer significant (*p* > 0.05). 

#### Negative Automatic Thoughts as a Specific Pathway to Experiential Avoidance and Cognitive Fusion

We next examined whether NAT represent a specific pathway to higher levels of EA, as well as CF. To investigate this, NAT were investigated as a mediator in the relation between group (individuals with schizophrenia vs. individuals with end-stage CKD) and EA (Model 1), as well as between group and CF (Model 2). As shown in Figure 1, NAT were a significant mediator between group and EA (B = 3.30, CI., 0.980, 0.560), as well as between group and CF (B = 3.05, CI., 0.814, 5.866). In addition, the direct effect of group on EA (B = 2.66, *p* > 0.05) and CF (B = 3.06, *p* > 0.05) was no longer significant, suggesting that NAT are a full mediator in these relations. In other words, having schizophrenia was related to facilitating more NAT, which in turn explained the higher levels of EA, as well as CF. However, when age and gender were entered as covariates, the mediation models were no longer significant (*p* > 0.05). 

## 4. Discussion

The present study explored the role of NAT, CF and EA in schizophrenia compared to individuals with end-stage CKD. First, an important outcome of our study was the increased intensity of NAT in the schizophrenia group, as compared to the end-stage CKD group. Previous investigations documented the impact of NAT in psychoses, pointing to the fact that schizophrenia patients present higher NAT than individuals without psychoses [56,57]. Similarly, it has been shown that during the acute phase, negative beliefs about self, others and future could be more active in schizophrenia, impacting the way patients evaluate various life events [58]. Along with frequency, we also assessed the credibility of NAT. Presumably, increased credibility of NAT could point to an important process in schizophrenia, which may contribute to patients’ interpretations of delusional content as real and threatening, which is associated with an exaggerated reaction towards internal experiences and a distorted perception of reality in these individuals. 

Second, schizophrenia patients showed higher levels of CF and EA, which is in line with previous studies [59,60], indicating that these two processes could be distinctive features between schizophrenia patients and the control group. For example, it has been shown that individuals diagnosed with a psychotic disorder frequently used EA for coping with challenging internal experiences [60]. At the same time, CF and EA have both been positively correlated to psychotic symptoms in schizophrenia [61]. In this way, our findings are also concordant with the conclusions of a systematic review stating that difficulties related to EA, namely the unwillingness to experience psychological distress, are more frequent among people with psychoses [62]. Notably, CF has been described as a dysfunctional process that may increase the likelihood of psychotic symptoms in people who are at high risk [63]. 

Third, a major finding of our investigation was that NAT were mediators in the relation between an existing schizophrenia diagnosis and CF. This is consistent with other results indicating that NAT and CF are highly correlated, yet different constructs [64]. More specifically, we tested a model in which the experience of NAT in individuals with schizophrenia may further explain the experience of CF, which describes individuals’ reaction to and degree of confidence in their negative thoughts. In addition, we found that NAT also mediated the pathway to EA in schizophrenia patients. Similar results have been obtained in studies emphasizing the connection between NAT and EA, pointing to the fact that both have an important role in generating distress [65,66]. 

The idea that NAT mediate the path to disturbing psychological symptoms was formulated within the cognitive theory and has been discussed especially in relation to clinical samples [13,66]. Likewise, we indicated that NAT operate as a primary process that may explain CF and EA in schizophrenia, which may further create a specific dysfunctional pattern involved in the exacerbation and maintenance of this psychiatric disorder. Our results are also in line with investigations that demonstrated an association between NAT, CF, and EA [65,66]. More precisely, we showed that NAT could exert a direct influence on EA in schizophrenia patients. This can be related to the fact that cognitive impairment has been proven to be a central characteristic of schizophrenia, contributing to its specific clinical manifestations [67].

Future research may seek to shed light on the relation between NAT and psychological inflexibility dimensions, as well as their differential involvement among individuals with schizophrenia and other types of disorders. It is important to note that when controlling for age and gender, no significant differences emerged. Thus, in the current study, we could not detect if the significant differences in NAT, EA, and CF were due to clinical differences between the groups (schizophrenia vs. end-stage CKD) or due to differences in age and gender. These biases can be related to the symptoms severity in individuals with schizophrenia and/or end-stage CKD and/or to the small sample size. In this context, future studies are needed to further augment current results.

### 4.1. Clinical Implications of This Study

Based on the results of this study, therapeutic approaches with the largest empirical support are recommended, including recovery-oriented cognitive therapy(CT-R) and cognitive behavioral therapy for psychosis (CBTp), which target the identification of NAT that accompany psychotic experiences and their reappraisal [68]. Our indication builds on multiple investigations of the effectiveness of various CBT approaches for the treatment of schizophrenia [69]. Furthermore, “third-wave” cognitive-behavioral therapies, such as acceptance and commitment therapy (ACT), apply mindfulness and acceptance-based methods for decreasing psychological inflexibility in schizophrenia patients [70]. Most of these interventions concentrate on NAT, CF, EA, and other cognitive biases, mobilizing change processes that can contribute to decreases in clinical symptoms [71]. All these interventions are complementary, targeting thought processes involved in schizophrenia with the purpose of reducing the level of PI. Additionally, for a better therapeutic result, it is recommended to combine antipsychotic treatment with cognitive-behavioral therapy. This combined treatment can be focused on two directions: (1) personalized antipsychotic medication targeting the dopamine system for decreasing cognitive deficits [72]; (2) cognitive-behavioral therapy addressing NAT and PI. This intervention can be considered a personalized therapeutic approach in schizophrenia, in which antipsychotic medication contributes to decreased cognitive impairments targeting the dopaminergic system, which may in turn improve the psychological interventions within CBT. Future studies should investigate the efficacy of this combined/personalized treatment.

### 4.2. Limitations

The present study has several limitations. First, the presence of other comorbidities such as depression and anxiety were not evaluated in either the target group or the control group; these could represent important aspects regarding specific characteristics and differences between the groups. However, this study focused on investigating transdiagnostic processes involved in the overall quality of life in patients with chronic somatic and psychiatric conditions. This could inform the implementation of psychotherapeutic interventions targeting the mechanisms involved in the occurrence of emotional disorders and not only the reduction of their symptoms. Further exploration of these comorbidities in both groups could provide a clearer picture regarding the particularities of anxious manifestations and depressive symptoms’ intensity in patients with schizophrenia and end-stage CKD respectively. Since the present study included an inpatient sample, the assessment of anxiety and depression was hindered by time constraints and patients’ desirability. Future research could evaluate the association between depression and anxiety and specific cognitive processes in schizophrenia. Second, the study did not include a clinical scale to measure the severity of specific schizophrenia symptoms in order to assess the relations between these symptoms, NAT, CF and EA. Future studies could investigate the relations of those cognitive processes with specific symptoms of schizophrenia to determine the implications of NAT, CF and EA in this psychiatric disorder. Third, patients had been taking pharmacological treatment for at least five years when the study was conducted, which may have influenced the way they responded to some of the psychological scales, therefore biasing the results.

## 5. Conclusions

High levels of NAT (both in terms of frequency and credibility), CF and EA could be considered specific dysfunctional cognitive processes in schizophrenia. Additionally, NAT play a mediating role between two important dimensions of PI, namely CF and EA, in schizophrenia. Even though the presence of a chronic medical condition and the presence of NAT and PI describe both between schizophrenia and end-stage CKD, the intensity of NAT, CF and EA is higher in individuals with schizophrenia. These outcomes have important implications for research on the psychological mechanisms involved in schizophrenia, as well as the psychotherapeutic approach used in the treatment of this severe psychiatric disorder.

## Figures and Tables

**Figure 1 jcm-11-00871-f001:**
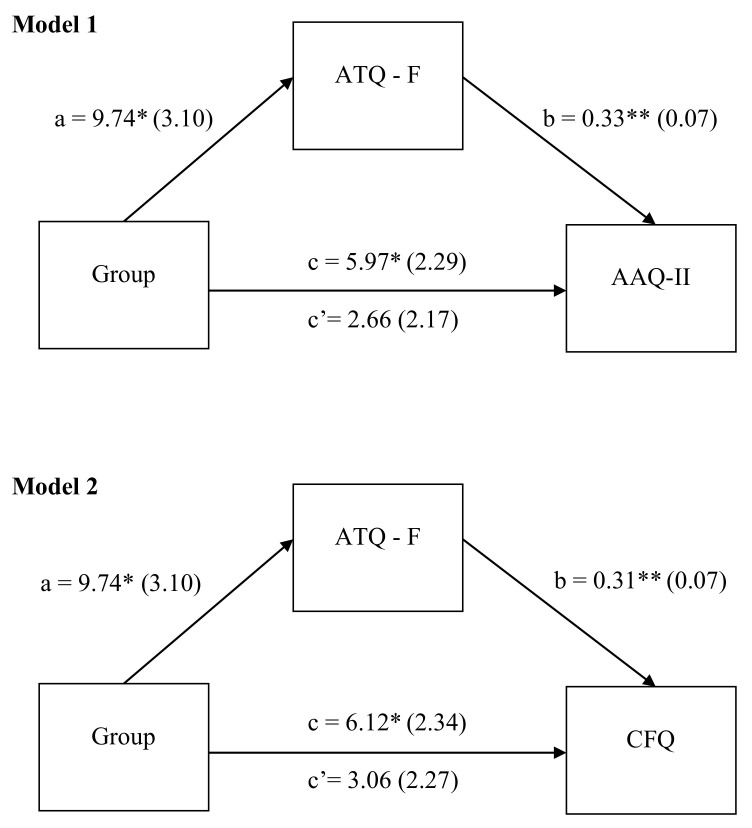
Negative automatic thoughts as a mediator between group (0 = individuals with end-stage CKD; 1 = individuals with schizophrenia) and experiential avoidance (Model 1), as well as between group (0 = individuals with end-stage CKD; 1 = individuals with schizophrenia) and cognitive fusion (Model 2). Pathways a, b, c and c’ represent unstandardized regression coefficients and standard errors. ATQ-F = the Automatic Thoughts Questionnaire-Frequency; AAQ-II = the Acceptance and Action Questionnaire II; CFQ = the Cognitive Fusion Questionnaire. Path a represents the associations between group and NAT. Path b represents the associations between NAT and CF. Path c represents the total association between group and CF. Path c’ represents the direct association between group and PI dimensions (EA and CF respectively). * *p* < 0.05; ** *p* < 0.001.

**Table 1 jcm-11-00871-t001:** Demographic characteristics of participants by group.

Sample Characteristics	Individuals with Schizophrenia (*n* = 41)	Individuals with End-Stage CKD (*n* = 40)	Statistical Significance (χ^2^/t)	*p*
Gender *n* (%) female, χ^2^	23 (63.9)	13 (36.1)	4.56	0.033
Age range, M (SD), t	23–67 44.98 (11.74)	44–75 60.38 (9.14)	6.57	0.001

M = mean; SD = standard deviation.

**Table 2 jcm-11-00871-t002:** Descriptive statistics for negative automatic thoughts, experiential avoidance, and cognitive fusion.

Measure	Individuals with Schizophrenia (*n* = 41) M (SD)	Individuals with End-Stage CKD (*n* = 40) M (SD)	Statistical Significance (F)	Partial η^2^
ATQ-F	38.02 (16.79)	28.27 (10.35)	F (1, 79) = 9.833; *p* = 0.002	0.111
ATQ-C	40.24 (17.95)	33.20 (11.72)	F (1, 79) = 4.348; *p* = 0.040	0.052
AAQ-II	25.04 (12.17)	19.07 (7.99)	F (1, 79) = 6.775; *p* = 0.011	0.079
CFQ	28.17 (12.07)	20.12 (6.87)	F (1, 79) = 6.834; *p* = 0.011	0.080

ATQ-F = the Automatic Thoughts Questionnaire - Frequency; ATQ-C = the Automatic Thoughts Questionnaire-Credibility; AAQ-II = the Acceptance and Action Questionnaire II; CFQ = the Cognitive Fusion Questionnaire; M = mean; SD = standard deviation.

**Table 3 jcm-11-00871-t003:** Zero-order correlations between the variables in the study for individuals with schizophrenia.

Variable	1	2	3	4
1. ATQ-F	-	0.833 **	0.511 **	0.396 *
2. ATQ-C		-	0.550 **	0.553 **
3. AAQ-II			-	0.801 **
4. CFQ				-

ATQ-F = the Automatic Thoughts Questionnaire-Frequency; ATQ-C = the Automatic Thoughts Questionnaire - Credibility; AAQ-II = the Acceptance and Action Questionnaire II; CFQ = the Cognitive Fusion Questionnaire; * *p* ≤ 0.05; ** *p* ≤ 0.001.

**Table 4 jcm-11-00871-t004:** Zero-order correlations between the variables in the study for individuals with end-stage CKD.

Variable	1	2	3	4
1. ATQ-F	-	0.420 **	0.329 *	0.466 **
2. ATQ-C		-	0.685 **	0.750 **
3. AAQ-II			-	0.815 **
4. CFQ				-

ATQ-F = the Automatic Thoughts Questionnaire-Frequency; ATQ-C = the Automatic Thoughts Questionnaire-Credibility; AAQ-II = the Acceptance and Action Questionnaire II; CFQ = the Cognitive Fusion Questionnaire; * *p* ≤ 0.05; ** *p* ≤ 0.001.

## Data Availability

The data presented in this study are available on request from the corresponding author (peter.olah@umfst.ro).

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
