# Peer review of "The Relation between Negative Automatic Thoughts and Psychological Inflexibility in Schizophrenia"

_jcm, 2022, doi:10.3390/jcm11030871_

Round 1

Reviewer 1 Report

Thank you for the opportunity to review this interesting manuscript. Overall, this paper has a relevant focus and is of interest for researchers and clinicians aiming at improvement of mental health in psychiatric population. I have some remarks, mainly ideas that in my opinion should be clarified. I hope the authors will find the comments helpful for the further process.

Introduction

Introduction is well developed regarding the main topics. However, for me, further justification on why authors choose to compare individuals with schizophrenia with individuals with Chronic Kidney Disease is needed. I understand that authors provided a short explanation on Methods section but, in my opinion this option should be elaborated. Which are the characteristics of the psychological process in individuals with Chronic Kidney Disease and why are they relevant to compare with psychiatric population? In this justification, comparison/contrasting literature is crucial. On the methods section, when authors provided a short explanation and stated some evidence about the characteristics of individuals with Chronic Kidney Disease, no references are provided.

Materials

When describing the instruments please be consistent with the included information: reference for original instrument, reference for translation, number of questions, type of scale, calculation, psychometric properties of original version and translated version and type of sample, should be present in all included instruments.

Limitations

Limitations need expanding in my view; More technical discussion of limitations could be given.

Specifically, regarding the second limitation: “the presence of other comorbidities such as depression and anxiety were not evaluated in neither the target group nor the control group, which could represent important aspects regarding specific characteristics and differences between the groups”; For me this is the major limitation of the present study. Due to the importance of depression and anxiety states in mental health and specially  on psychological processes, more technical discussion should be added.

Reviewer 2 Report

In this manuscript, Popa and colleagues report the relation between negative automatic thoughts and psychological inflexibility in schizophrenia. The results are interesting. However, some minor points should be addressed before publication.

  • Authors need to better introduce and discuss cognitive deficits in schizophrenia. In particular it should be remarked that cognitive deficits are the most clinically relevant dimension of the disease, leading to significant distress or impairment in social, occupational, or other important areas of functioning.

  • Individuals with schizophrenia are basically treated with antipsychotics. In this respect, The Authors report that “patients were taking pharmacological treatment for at least five years when the study was conducted, which may influence the way they responded to some of the psychological scales, therefore, biasing the final results”. In this respect, the Authors might want to discuss that better pharmacological treatments are also needed. Indeed it has been recently reviewed (please report the following reference: PMID: 33167370)  the fact that  more personalized treatments with antipsychotics may have beneficial effects on cognitive dysfunctions. Thus, can a better use of drugs make the psychological interventions more effective?
